# From iPSCs to Pancreatic β Cells: Unveiling Molecular Pathways and Enhancements with Vitamin C and Retinoic Acid in Diabetes Research

**DOI:** 10.3390/ijms25179654

**Published:** 2024-09-06

**Authors:** Felipe Arroyave, Yomaira Uscátegui, Fernando Lizcano

**Affiliations:** 1Center of Biomedical Investigation (CIBUS), Universidad de La Sabana, Chia 250008, Colombia; 2Doctoral Program in Biociencias, Universidad de La Sabana, Chia 250008, Colombia; 3School of Medicine, Universidad de La Sabana, Chia 250008, Colombia

**Keywords:** islet beta cells, differentiation, iPSC, insulin, diabetes, cell therapy

## Abstract

Diabetes mellitus, a chronic and non-transmissible disease, triggers a wide range of micro- and macrovascular complications. The differentiation of pancreatic β-like cells (PβLCs) from induced pluripotent stem cells (iPSCs) offers a promising avenue for regenerative medicine aimed at treating diabetes. Current differentiation protocols strive to emulate pancreatic embryonic development by utilizing cytokines and small molecules at specific doses to activate and inhibit distinct molecular signaling pathways, directing the differentiation of iPSCs into pancreatic β cells. Despite significant progress and improved protocols, the full spectrum of molecular signaling pathways governing pancreatic development and the physiological characteristics of the differentiated cells are not yet fully understood. Here, we report a specific combination of cofactors and small molecules that successfully differentiate iPSCs into PβLCs. Our protocol has shown to be effective, with the resulting cells exhibiting key functional properties of pancreatic β cells, including the expression of crucial molecular markers (pdx1, nkx6.1, ngn3) and the capability to secrete insulin in response to glucose. Furthermore, the addition of vitamin C and retinoic acid in the final stages of differentiation led to the overexpression of specific β cell genes.

## 1. Introduction

Diabetes mellitus (DM) is a chronic, non-transmissible disease recognized as a leading cause of mortality globally, with high morbidity due to the chronic deterioration of insulin-producing cells [1]. It is estimated that diabetes affects one in ten individuals worldwide, with a projected global-specific health expenditure that rapidly approaches USD 1 trillion [2,3]. Furthermore, diabetes induces a series of vascular complications caused by different inflammatory, metabolic, and oxidation stress mechanisms [4,5,6]. Diabetes primarily manifests itself as type 1 diabetes, in which an autoimmune process destroys insulin-secreting cells [7,8], and type 2 diabetes mellitus, characterized by the progressive destruction of islets due to functional alteration of insulin [9]. Although medical therapy can delay complications in T2DM, curing diabetes is a desirable objective worldwide [10]. Given these challenges, regenerative medicine presents a promising solution. Recent years have seen remarkable progress in organ and tissue regeneration in vitro [11,12,13]. Endocrine pancreatic tissue regeneration has demonstrated encouraging results, restoring carbohydrate metabolism in primate models and patients with DM [14,15].

Pluripotent stem cells (PSCs) have been proposed as an encouraging medical treatment due to their high plasticity and multiple sources. PSCs can be obtained from embryonic tissues, umbilical cords, and some cellular “niches” in adult tissue [16]. However, these sources of pluripotent cells may have ethical limitations or be restricted and difficult to extract. Reprogramming somatic cells into induced pluripotent stem cells (iPSCs) offers an alternative, increasing the limited supply of pluripotent cells extracted from adult tissue [17]. These cells can be derived from the patients or compatible donors and differentiated to replace affected tissue, thus avoiding immunological rejection and immunosuppressive therapies [18,19].

Several methods to differentiate β-pancreatic cells have been developed recently, utilizing various cofactors and small molecules to replicate pancreatic development in vitro. Molecules like activin A, fibroblast growth factor (bFGF), fibroblast growth factor-10 recombinant protein (FGF10), NOGGIN, and SANT-1 have stimulated the activation of different molecular pathways involved in pancreatic differentiation. However, many questions remain regarding how pancreatic development can be successfully achieved in vitro and how the optimal combination of molecules and cofactors, especially vitamin C and retinoic acid, guarantees a substantial number of functional β-cells [20,21,22,23].

Vitamin C and retinoic acid play a crucial role during pancreatic differentiation protocols. In particular, the addition of these two molecules has been reported to favor the differentiation of insulin-producing cells from induced pluripotent cells.

Generally, vitamin C induces epigenetic stabilization and favors transcription due to its crucial role as a cofactor during the activation and repression of specific genes during differentiation. Similarly, high concentrations of vitamin C have been shown to reduce oxidative stress by protecting cells from oxidative damage during developmental protocols. For its part, retinoic acid acts through receptors that modulate the expression of characteristic genes during differentiation. Likewise, it has been reported that high concentrations of retinoic acid favor the formation of pancreatic structures similar to those of the adult pancreas. This allows for maintaining the cellular identity and avoiding the detour towards other undesired lineages, such as the hepatic one [24,25].

The widespread nature of DM necessitates access to preventive and therapeutic resources for many patients. Current treatment options are limited, particularly for those with diminished beta cell numbers or complete functional decline, leading to the high cost of chronic treatment. This study aims to introduce a comprehensive protocol for the in vitro generation of insulin-producing pancreatic β cells. Further research into the molecular physiology of beta cells and insulin secretion is crucial for developing effective treatment strategies. We aim to establish a protocol that effectively produces mature insulin-secreting beta cells widely available at low costs.

## 2. Results

### 2.1. iPSCs Growth and Maintenance

Maintenance of pluripotency before the differentiation assay is critical to guarantee the protocol’s good performance. After 72 h of culture in a complete mTeSR1 medium, the cells showed the expected [26] growth characteristics, such as typical growth with tightly packaged and distinct borders (Figure 1). The growth characteristics obtained before the initiation of the differentiation assay were adequate according to previously reported protocols [26,27].

Figure 2 shows the expected growth characteristics of iPSC morphology, which differentiates iPSCs into PβLCs. The colonies grow in tightly packaged clusters with a high nuclear-to-cytoplasm ratio and distinct borders. Images were captured with a phase contrast microscope (ZEISS AX10, Carl Zeiss, Oberkochen, Germany).

### 2.2. Differentiation of Definitive Endoderm from iPSCs

This study verified the effects of molecules such as (activin A, CHIR99021, bFGF, FGF10, EGF, SANT-1, retinoic acid, and ascorbic acid) during pancreatic in vitro development. These cytokines and small molecules are essential in activating and repressing signaling pathways such as Wnt, activin A, and hedgehog, which are involved in pancreatic differentiation.

The strategy to generate PβLC in vitro from iPSCs is outlined in material and methods. The basal medium was mTeSR1 without specific medium supplementation; the growth medium was only supplemented with cytokines and small molecules modified from Korytnikov and Nostro [21]. The protocol requires two weeks and is developed as a sequence in which each stage represents a different differentiation period. During each step, the mixture of cofactors and small molecules orchestrates the differentiation of iPSCs into PβLC. The differentiation is developed under serum-free/xeno-free (SF-XF) conditions to avoid any external cofactor that may interfere with the results, such as the fetal bovine serum (FBS) [28,29,30].

Some selected molecules for the development study have been previously reported responsible for endodermal differentiation, especially toward pancreas development [20,21,31,32,33]. The protocol allowed us to determine the proteins and genes expressed at each pancreatic differentiation stage. The gene expression and protein results (Figure 3 and Figure 4) showed that this methodology activated a process to obtain, in the beginning, cells of the definitive endoderm and, subsequently, PβLC that express characteristic pancreatic β cell markers. Notably, at the end of our protocol, only insulin expression and no other endocrine hormones, such as somatostatin (*sst*) and glucagon (*gcg*) (detected by qPCR) (Figure 3E), were obtained, showing that the cytokines and small molecules were adequate to ensure the production of PβLCs. Our results were confirmed by comparing protein expression between our differentiated cells (stage 4) and the insulin-secreting cells (Min6).

The generation of PβLCs was performed with the initial production of DE by incubating iPSCs with a mixture of activin A (100 ng/mL) and CHIR99021 (2 µM/mL) for 24 h. As previously reported, the iPSC cell-derived DE expressed the *sox17* and *foxA2* markers (Figure 3A and Figure 4A). Next, CHIR99021 was removed from the differentiation medium, and activin A (100 ng/mL) was used for another 24 h with bFGF and ascorbic acid, which allowed the DE-differentiated cells to maintain the lineage. The activin A concentration selected for stage 1 of differentiation was the same as that applied during stage 0 (100 ng/mL). We opted for this concentration due to the potential impact of lower levels of activin A during later stages of endodermal differentiation on the eventual yield of functional cells [32]. We also used a low concentration of CHIR99021 (2 µM/mL), which has been shown to induce endodermal differentiation. Otherwise, higher doses of CHIR99021 (5 mM/mL) can generate mesodermal differentiation [33,34,35,36].

### 2.3. Differentiation of Definitive Endoderm into Pancreatic Progenitors

After DE was achieved, pancreatic progenitors were observed using bFGF, ascorbic acid, and FGF10 molecules. The expression of the *hnf4A* and *sox9* (Figure 3B,C and Figure 4A) demonstrated the maintenance of pancreatic identity during differentiation. In the present protocol, CHIR99021 was removed after 24 h and replaced with ascorbic acid. This step allowed us to obtain a strong expression of *hnf4A* and the continued expression of *sox17* and *foxA2* [20,28,29]. In addition, the concentration and exposure time of FGF10 used during our protocol conferred the cells an endocrine identity determined by the later expression of *pdx1* (subsequently detected) mediated by the reduction of the NOTCH signaling pathway (restricted to endocrine differentiation). Maintaining a PP differentiation stage is demonstrated by the later expression of characteristic genes such as *ptf1a* (Figure 3D) [37,38,39].

Notably, our results demonstrate a process of pancreatic differentiation confirmed by the expression of *sox9* during the stage 3 protocol (Figure 3C). *sox9* is a characteristic marker of pancreatic differentiation. This molecule is exclusively expressed by the endocrine precursor cells during the expansion of the pancreatic epithelium, favoring the initial expression of *pdx1*, as observed in the results (Figure 3C,D) [40]. 

Pancreatic differentiation from DE to PP is closely related to the activation and suppression of the NOTCH, Wnt, BMP, and FGF signaling pathways due to the expression of the *sox9* and *pdx1* genes related to pancreatic differentiation (Figure 3C–E). However, during experimentation, hepatic differentiation genes such as *hnf4A* and *foxA2* expression suggested a close association between the pancreatic and hepatic progenitors (Figure 3C,D) [41].

### 2.4. Differentiation of Pancreatic Progenitors into Pancreatic Beta-like Cells

After obtaining PP, our protocol used a mix of cofactors and small molecules to minimize the expression of signaling pathways such as NOTCH, bone morphogenic proteins (BMP), and transforming growth factor beta (TGF-β) [25]. The cofactors and small molecules used during the S3 and the S4 allowed the expression of pancreatic β cell characteristic genes such as *ngn3*, *pdx1*, *nkx6.1*, and ins (insulin) [42] (Figure 3 and Figure 4). However, it is noteworthy that signaling pathways such as NOTCH, BMP, and TGF-β are essential at the beginning of the pancreatic differentiation, where molecules such as activin A, glutamine, and CHIR99021 are used to favor and block their expression to allow hiPSCs to differentiate to a DE state. Nevertheless, with the development of the present protocol, those signaling pathways are suppressed, especially the BMP, because it is highly associated with hepatic differentiation. Therefore, we used Noggin (a BMP inhibitor) and SANT-1 to promote pancreas specification and stop the cells of a hepatic lineage differentiation [43].

The final step was to develop PβLC from PP. Notably, the *ngn3* gene must be expressed at the end of the protocol and not in other stages to avoid the differentiation of polyhormonal cells (alpha, gamma, etc.) (Figure 3). Early expression of the *ngn3* gene will result in many polyhormonal cells [44]. As previously reported, our protocol used vitamin C at this differentiation stage to prevent polyhormonal cell formation [43]. 

During the last step of our protocol, our research group decided to add retinoic acid again due to its considerable importance in increasing the expression of *pdx1*, which favors the differentiation and maturation of pancreatic β cells. However, to guarantee the success of this process, half the concentration of retinoic acid was used in the S4 medium compared to the S3 medium because using low concentrations of this cofactor during the final stages of differentiation is crucial to maintain constant *pdx1* expression (Figure 3C–E) and favor the initiation of *nkx6.1* gene expression (Figure 3E), which seems to be restricted exclusively in pancreatic β cells (Figure 4) [39,45,46].

### 2.5. Glucose-Stimulate Insulin Secretion

PβLCs underwent a glucose challenge to assess their glucose-stimulated insulin secretion (GSIS) capability. Figure 5 shows the results obtained from the insulin Elisa test for the PβLC, the insulin-secreting cells (Min6), and two negative controls (iPSCs w/o differentiation and HEK293T cell line).

The differentiated cells’ insulin secretion showed the effectiveness of the protocol developed by our group. While the insulin levels from PβLC were not as high as those observed from the insulin-secreting cells, their capacity to respond to glucose by measurably secreting insulin in vitro suggests potential functionality. However, further research is essential to fully determine the differentiated cells’ maturation status and functionality in future rodent models. Generally, as previously reported by insulin-producing *pdx1^+^/nkx6.1^+^* cells, they will show functionality and maturation during in vivo graft experiments. 

## 3. Discussion

In the current study, iPSC cells were differentiated into insulin-producing pancreatic β-like cells using small molecules and cofactors that mimicked embryonic development through activation/inhibition of cell-specific signaling [47]. The insulin secretion by the PβLCs was subject to glucose concentration in the medium, making it a good indicator of real insulin-producing beta-like cells. Our protocol is based on previous protocols, with some modifications to reduce the effect of uncontrolled external factors and to stimulate the differentiation of insulin-producing cells [48,49,50].

The morphological changes detected during the protocol demonstrate that the combined cofactors and small molecules selected during the study effectively differentiated into iPSCs. However, some apoptotic cells are present during the differentiation protocol, likely due to changes in mitochondrial function triggering typical apoptotic behavior [51]. 

During differentiation, apoptosis in iPSCs is mainly attributed to p53 mitochondrial translocations, cytochrome c release, and reactive oxygen species (ROS) production. Figure 2 illustrates a reduction in apoptotic cells by the end of the protocol. This reduction may result from the maturation process, which decreases the cell proliferation rate and regulates p53, thereby inhibiting apoptosis [52,53]. 

Initially, iPSCs must differentiate toward definitive endoderm (DE), with sox17 and foxa2 being relevant markers in this early state. Activation of the Wnt pathway and TGF is crucial for this differentiation process. Various concentrations of activin A and CHIR99021 have been employed. In the current protocol, we observed that using activin A for 72 h at a concentration of 2 μM/mL and 100 ng/mL of CHIR99021 during 24 h was suitable for expressing DE markers (Figure 3A and Figure 4A) [43,54,55].

After obtaining the endodermal lineage, we directed the cells to the pancreatic gut tube (PG) (stage 1 of our differentiation protocol), which is considered the period in which the cell fate is determined to be liver or pancreatic lineage, so the mixture of cofactors and small molecules should be used appropriately [43,47,56]. The induction of the pancreatic lineage was performed by bFGF 5 ng/mL, a cofactor responsible for the development of the pancreatic epithelium [38]. It is important to consider that lower doses of FGF have been used for hepatic–hepatic-like differentiation [57]. In this stage, the expression of *hnf4a* and the preservation of *foxA2* and *sox17* expression are good indicators of pancreatic gut differentiation (Figure 3A,B) [20,29].

FGF10 and dorsomorphin are added to develop the pancreatic lineage, which initiates the unfolding of pancreatic epithelium. At this stage, pdx1 and sox9 are reliable indicators of pancreatic progenitors (Figure 3C and Figure 4A) associated with the definitive establishment of the pancreatic lineage. The presence of the transcription factor pdx1 manifests the identity of the future cells [58], which will later transform into insulin-producing β cells. FGF 10 is a fibroblast growth factor that induces epithelial branching, and the addition of B27 as a supplement increases the development of the cells. We decided to maintain the use of B27 to promote and preserve the viability of our cells during the assay and reduce the stress and possible apoptosis caused by the continuous change of the medium [59,60]. FGF10 significantly impacts differentiation since it helps develop the pancreatic epithelium, restricting premature endocrine differentiation and maintaining the progenitor state of the cells through the activation of the notch signaling pathway [61]. Sox9 is essential in specifying and maintaining pancreatic progenitor cells and differentiation toward the beta cell lineage, which later differentiate into endocrine progenitors expressing pdx1, ngn3, and ptf1a [62,63]. Moreover, the cells’ expression of the sox9 gene during the differentiation protocol is a clear signal of the expansion of the pancreatic epithelium [40].

Subsequently, we achieved endocrine precursor cells characterized by the expression of markers such as *ptf1A* (Figure 3D). It is essential to highlight the use of cofactors such as FGF10, SANT-1, ascorbic acid, retinoic acid, and Noggin, which contribute to establishing pancreatic β identity during development. Notably, the continued use of FGF10 is needed during these final stages (PP and EP) because a deficiency of this cofactor can trigger hepatic differentiation [63,64,65]. Noggin (BMP inhibitor) and SANT-1 (inhibitor of sonic hedgehog (signaling) also prevented hepatic differentiation [48,66]. Likewise, ascorbic and retinoic acids became fundamental during this step, contributing to cell identity’s final establishment. Ascorbic acid prevents the formation of polyhormonal cells, while retinoic acid acts as an activator of the *pdx1* gene, increasing its expression and thus favoring future insulin expression [43]. Ascorbic acid was earlier used (Figure 1) to reduce the expression of ngn3 and thus prevent the appearance of acinar cells [67].

Finally, we obtained PβLCs from EP due to cofactors such as hEGF, nicotinamide, ascorbic acid, and retinoic acid. During this last step of our protocol, the production of insulin and the absence of other endocrine hormones, such as SST and GCG (Figure 3E), may be considered a positive result due to the absence of polyhormonal cells. Once differentiated cells obtain pancreatic identity, retinoic acid is crucial due to the production of diverse retinoic acid derivates produced by the enzyme retinaldehyde dehydrogenase type II [68] that promotes pancreatic development at an early stage [69] and later supports beta cell differentiation [70,71].

The determination of *nkx6.1* expression (Figure 3E) indicates the progression of a pancreatic progenitor state to a PβLC. *nkx6.1* is considered a transcription factor exclusively expressed by adult β-pancreatic cells (Figure 3E and Figure 4B) because it is directly related to the glucose metabolism of the Glut 2 gene [58].

The last step of our protocol was characterized by the effects of hEGF, which promoted the expansion of pancreatic cells, improving the number of *pdx1-positive* cells [72]. Likewise, using nicotinamide and Noggin induces the differentiation and maturation of cells that coexpress *pdx1/nkx6.1* [73,74], which may subsequently mature into functional β-cells [75] (Figure 3E and Figure 4B).

In this study, we employed a protocol utilizing two essential compounds: ascorbic acid, which has been previously reported as a critical compound in insulin release due to the effect associated with the potential modification of the conductivity of ion channels, and retinoic acid, to enhance the differentiation of pancreatic beta cells and, which, in turn, affects glucose metabolism and cellular uptake [20,21,31,76,77]. The rationale behind ascorbic acid and retinoic acid is their epigenetic influence, particularly their ability to increase gene expression through activating DNA demethylation proteins, such as the TET (ten-eleven translocation) enzymes [78,79]. Ascorbic acid is believed to significantly impact the differentiation and maintenance of pancreatic β-cell identity by influencing insulin release. The ascorbic acid addition at the same chemical concentration at the end of our protocol allowed the PβLC obtained during our study to achieve the expression of insulin at considerable levels compared to the concentration obtained in previous studies by Davis et al. (2020) [42].

Davies and collaborators showed that their differentiated cells produced insulin. However, the insulin levels obtained did not achieve the levels obtained by the cadaveric islets used as their positive control. Likewise, their study showed that high levels of glucose during the GSIS (16.7 mM) resulted in insulin secretion inhibition. In contrast, during our study, levels of 20 mM of glucose during the GSIS showed increased insulin release. Nevertheless, the insulin expression of the PβLC was not similar and was under the insulin produced by the insulin-secreting cells used during this study (Figure 5) [42,80]. 

During our study, we reintroduced retinoic acid at the final stage of differentiation due to its crucial role in regulating the expression of the transcription factor Pdx1, which is essential for pancreatic development. However, it was necessary to adjust the concentration of retinoic acid carefully [67,81]. While it benefits pancreatic differentiation, an excessive amount can inhibit endocrine differentiation and disrupt insulin expression [82,83]. Therefore, maintaining an optimal concentration of retinoic acid is vital to promote pancreatic differentiation and β-cell maturation while preventing the formation of undesired cell phenotypes within the pancreatic islets [25,84].

At the end of our study, PβLC were subjected to a glucose challenge to demonstrate their responsiveness to different glucose concentrations. Initially, we measured basal insulin using low glucose concentrations in KrB buffer and subsequently measured stimulated insulin using high glucose concentrations (20 mM). Although our results demonstrate insulin production in response to glucose (Figure 5), they do not reach the same production levels as the insulin-producing cell. Likewise, the levels of insulin released by the PβLC are below those obtained previously [85].

Lee et al. obtained insulin levels higher than 60 uIU/mL after 12 days of the protocol with concentrations between 20–30 mM glucose, while our cells produced a maximum of 8 uIU/mL after 14 days of differentiation. This difference in insulin production may be associated with the differences found between the protocols developed by the two groups [85,86].

## 4. Materials and Methods

### 4.1. Materials

The 20b iPSCs were acquired from the Harvard Stem Cell Institute (HSCI) (Boston, MA, USA). Vitronectin (VTN-N; A14700) recombinant human protein, truncated, Dulbecco’s phosphate-buffered saline without calcium and magnesium (DPBS; 14190250), L-Glutamine (25030081), activin A recombinant human protein (PHC9564), heat-stable recombinant human basic bFGF, heat stable FGF10 (PHG0375), B27 supplement minus vitamin A (12587010), human Noggin recombinant protein (PHC1506), human epidermal growth factor recombinant protein (EGF), Dulbecco´s modified Eagle’s medium (DMEM) plus GlutaMax10569010) and, fetal bovine serum (FBS) (A5256701) were obtained from Thermo Scientific, Waltham, MA, USA; mTeSR1 (85850) medium was acquired from StemCell Technologies, Vancouver, BC, Canada. CHIR99021 (SML1046), L-Ascorbic acid (A4544), dorsomorphin (P5499), nicotinamide (N0636), retinoic acid (R2625), monothioglycerol (MTG6145) and penicillin/streptomycin (P4333) were obtained from Sigma-Aldrich, St. Louis, MO, USA; SANT-1 (559303-5MG) was acquired from Merck-Millipore, Burlington, MA, USA.

Mammalian protein extraction reagent (M-PER 78501), Halt protease, phosphatase inhibitor cocktail, and Tween™ 20 Surfact-Amps™ detergent solution were obtained from Thermo Scientific, United States. A bicinchoninic acid kit (BCA) and Laemmli sample buffer (61-0737) were acquired from (Bio-Rad, St. Louis, MO, USA). Primary antibodies against forkhead box A2 (FOXA2) (710730), SRY-box transcription factor 17 (SOX17) (PA5-72815), hepatocyte nuclear factor alpha (HNF4A) (PA5-82159), SRY-box transcription factor 9 (SOX9) (PA5-81966), pancreas-associated transcription factor 1a (PTF1A) (PA5-112677), and Neurogenin 3 (NeuroG3) (703206), goat anti-rabbit IgG (H+L), secondary antibodies, and horseradish peroxidase conjugate (G21234) were obtained from Invitrogen, Waltham, MA, USA. A recombinant pancreatic and duodenal homeobox 1 (PDX1) (ab219207) and recombinant NK6.1 homeobox 1 (NKX6.1) antibodies (ab221549) were obtained from Abcam, United Kingdom. Immobilon-P PVDF membrane (IPVH00010) and Luminata Crescendo Western HRP Substrate were obtained from Merck Millipore, Darmstadt, Germany. A QuickExtract™ RNA extraction kit (QER090150) was obtained from Biosearch Technologies, Hoddesdon, UK), and OneScript^®^ Plus Reverse Transcriptase (G237) was obtained from ABM, Richmond, BC, Canada.

### 4.2. iPSC Culture and Maintenance

The 20b iPSCs were cultured in 6-well plates coated with vitronectin and mTeSR1 medium. The plates were incubated at 37 °C and 5% CO_2_ during all expansion and differentiation stages. Min6 cells (mouse insulinoma six cells) were kindly donated and used as an insulin-secreting result for β-cell differentiation [87]. Min6 cells were cultured in DMEM with 20% fetal bovine serum, 1% Ciprofloxacin (Sigma-Aldrich, St Louis, MO, USA), and 1% penicillin/streptomycin. All the plates were incubated at 37 °C and 5% CO_2_ during all the cultures, including expansion. Protein extraction was performed on Min6 cells. 

The iPSC culture was fed following this procedure daily: the culture media was aspirated, and 2 mL of fresh mTeSR1 medium per well was added. The culture medium was changed, and the cells were washed with 2 mL of DPBS until they reached a growth confluency of 85–90%. Once the iPSC culture was 85–90% confluent, the cells were transferred to new pre-coated vitronectin 12-well plates. 

### 4.3. Differentiation of iPSCs

The differentiation protocol was defined in 5 stages (Figure 6). In each stage, the mTeSR1 medium was supplemented with a diverse mixture of cytokines and small molecules to achieve a step-by-step differentiation process. Each step was carried out in triplicate under the same incubation conditions at 37 °C and 5% CO_2_.

Standard feed differentiation medium (SFD) and diluted MTG medium were prepared according to Korytnikov and Nostro [21]. Once a stage was performed, protein extraction, total RNA extraction, Western blot assays, and RT-PCR were carried out. Different stages were performed as follows:(1)Cells in 12-well plates were cultured until 85–90% confluency. mTeSR1 was changed daily, while DPBS w/o Ca++ and Mg++ was used to wash the wells [20,21].(2)Day 0: Stage 0 medium (S0) was supplemented with glutamine (1%), CHIR99021 (2 µM, activin A (100 ng/mL), and diluted MTG (3 µL/mL). The cells were incubated with an S0 medium for 24 h.(3)Days 1–2: The S1 medium was removed, and the cells were washed twice with DPBS w/o Ca++ and Mg++. S1 medium was supplemented with glutamine (1%), activin A (100 ng/mL), diluted MTG (3 µL/mL), heat-stable recombinant human basic fibroblast growth factor (bFGF) (5 ng/mL), and ascorbic acid (50 µg/mL). The medium was changed each day during stage 1.(4)Days 3–5: The S2 medium was removed, and the cells were rewashed with DPBS w/o Ca++ and Mg++ twice. S2 medium was supplemented with glutamine (1%), diluted MTG (3 µL/mL), dorsomorphin (0.75 µM), human fibroblastic growth factor-10 (FGF10) (50 ng/mL), and B27 supplement minus vitamin A (1%). Subsequently, the cells were incubated with the S2 medium, and the medium was changed each day during stage 2.(5)Days 6–7: The S3 medium was removed, and the cells were washed twice. S3 medium was supplemented with glutamine (1%), ascorbic acid (50 µg/mL), FGF10 (50 ng/mL), B27 (1%), SANT-1 (0,25 µM), retinoic acid (2 µM) and recombinant human noggin (Noggin) (50 ng/mL). Cells during stage 3 were incubated at 37 °C, 5% and CO_2_, and the medium was changed daily.(6)Finally, on days 8–12, the cells were cultivated with S4 medium supplemented with glutamine (1%), ascorbic acid (50 µg/mL), B27 (1%), Noggin (50 ng/mL), recombinant human epidermal growth factor (hEGF) (100 ng/mL), retinoic acid (1 µM), and nicotinamide (10 mM).

### 4.4. Western Blot

Once each differentiation stage was completed, the differentiation medium was collected, and the cells were washed twice with DPBS. Subsequently, the cells were homogenized in a mammalian protein extraction buffer (M-PER, ThermoScientific, Waltham, MA, USA) with a protease and phosphatase inhibitor cocktail (Halt, ThermoScientific, Waltham, MA, USA). The bicinchoninic acid assay (BCA) was used to determine the extracted protein content. Differentiation-related protein analysis was carried out using SDS electrophoresis and Western blotting. Briefly, equal protein amounts (20–30 µg/mL) were mixed with an equal volume of Laemmli sample buffer, denatured at 70 °C for 10 min, separated by 12% SDS-PAGE with a constant voltage of 100 V for 2 h, and then transferred to a PVDF membrane. The membrane was incubated in blocking buffer (TBST 1X + Tween-20 (0.1% + 5% skim milk powder) for 1 h at room temperature. The membrane was then incubated overnight at 4 °C with primary antibodies against SOX17 (5 µg/mL), FOXA2 (4 µg/mL), HNF4A (0.4 µg/mL), recombinant anti-PDX1 antibody (5:500) and recombinant anti-NKX6.1 antibody (5:500), SOX9 (0.4 µg/mL), PTF1A (1:5000), and NGN3 (1:1000). Subsequently, the membranes were washed three times with 1× TBST and incubated with HRP goat anti-rabbit IgG (H+L) secondary antibody (1:5000) for 2 h at room temperature. Finally, the membrane was exposed for 5 min at room temperature to Luminata™ Crescendo Western HRP Substrate and visualized by chemiluminescence using a myECL imager (ThermoScientific, Waltham, MA, USA).

### 4.5. Total RNA Extraction RT-PCR and Quantitative PCR (qPCR)

Total RNA extraction was performed using a QuickExtract™ RNA extraction kit according to the manufacturer´s protocol. Total RNA was quantified using a NanoDrop system (Invitrogen, Waltham, MA, USA). The cDNA was obtained using OneScript^®^ Plus Reverse transcriptase and oligo (dT) primers. qPCR was performed in triplicate on a CFX Opus 96 Real-Time PCR System (Bio-Rad, Hercules, CA, USA) with Luna^®^ Universal qPCR Master Mix (M3003S) (New England Biolabs, Ipswich, MA, USA). Expression data were normalized relative to the B-Actin transcript level. The fold change for each gene was calculated using the 2^−ΔΔCt^ method. The qPCR conditions were as follows: initial denaturation at 95 °C for 1 min followed by 40 cycles of 15 s at 95 °C, 30 s at 60 °C, and 15 s at 72 °C. Primer sequences are shown in Table 1.

The qPCR conditions were as follows: initial denaturation at 95 °C for 1 min followed by 40 cycles of 5 s at 95 °C, 10 s at 60 °C, and 15 s at 72 °C. Primer sequences are shown in Table 1.

### 4.6. Glucose Stimulation Insulin Secretion (GSIS)

Human insulin levels were determined using the Abcam Human Insulin ELISA kit (AB100578), Cambridge, UK. In summary, pancreatic beta-like cells (PβLC) underwent a double Krebs-Ringer buffer (Krb) wash and a 2-h pre-incubation in 2.5 mM glucose Krb. This was followed by a 1-h incubation in low-glucose Krb (2 mM) and supernatant collection. After another double wash in Krb, the cells were exposed to high-glucose Krb (20 mM) for 1 h, with subsequent supernatant collection. All measurements were carried out in triplicate.

### 4.7. Statistical Analysis

All the experiments were carried out at least three times. Data were displayed as mean ± SD. A two-tailed, unpaired Student’s *t*-test was performed to assess statistical significance, and *p* < 0.05 was considered significant.

## 5. Conclusions

In the present protocol, we successfully produced insulin-producing beta cells derived from iPSCs through four differentiation stages. We believe that the epigenetic regulation induced by ascorbic acid and retinoic acid plays a critical role in the differentiation of beta cells from iPSCs. A significant challenge in beta cell differentiation protocols is simplifying the production of mature beta cells and making them affordable for the general population worldwide. We consider that epigenetic regulation may play a principal role in these differentiation protocols, and it is possible that other small molecules can further enhance the differentiation of beta cells. Various protocols are currently under development, but the main goal remains to obtain pancreatic beta cells that are functionally active and capable of sustained insulin release in response to physiological cues. 

## Figures and Tables

**Figure 1 ijms-25-09654-f001:**
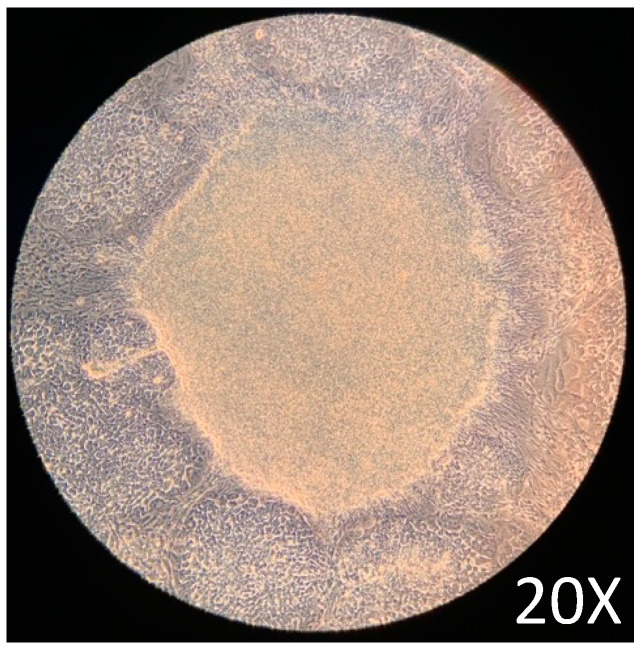
72 h iPSCs growth in complete mTeSR1 medium.

**Figure 2 ijms-25-09654-f002:**
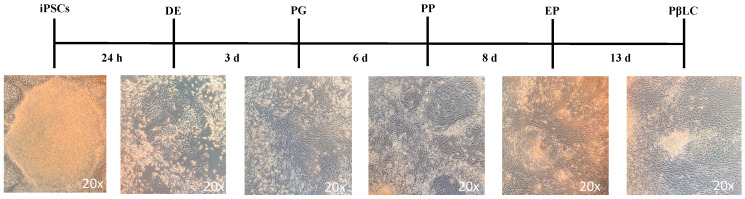
Schematic representation of the protocol to differentiate iPSCs into PβLCs. iPSCs were seeded on vitronectin-coated cell wells and incubated with the cofactors and small molecules indicated in the material and methods in a sequential protocol for 13 days. Images were captured with a phase contrast microscope (ZEISS AX10). Abbreviations: DE: Definitive endoderm; PG: Pancreatic gut tube; PP: Pancreatic progenitor; EP: Endocrine progenitor; PβLC: Pancreatic beta-like cell. h: hours, d: days.

**Figure 3 ijms-25-09654-f003:**
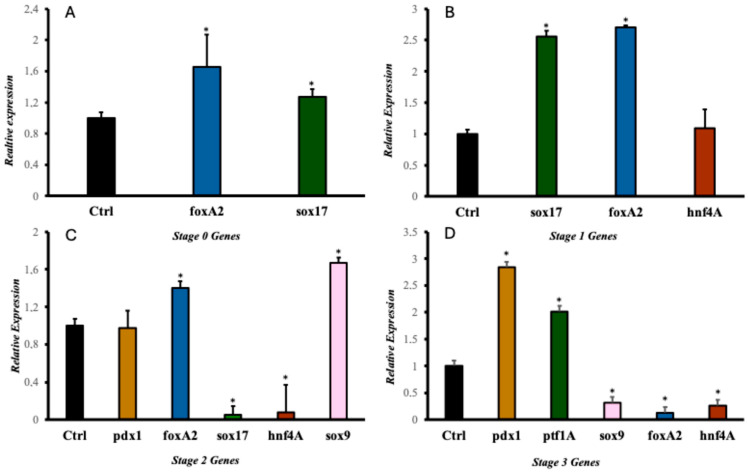
Gene expression during the differentiation process. (**A**) Stage 0 Gene expression (DE); (**B**) Stage 1 Gene expression (PG); (**C**) Stage 2 Gene expression (PP); (**D**) Stage 3 Gene expression (EP); (**E**) Stage 4 Gene expression (PβLC); (**F**) Insulin-producing cells gene expression. Expression data are normalized to the Ctrl (b-actin, transcript level). Each experiment was performed in triplicate. Bars under the same symbol (*) are statistically different under the two-tailed, unpaired Student’s *t*-test compared to the Ctrl expression level ** p* < 0.05; *** p* < 0.01. Abbreviations: *foxA2*: Forkhead box A2; *sox17*: SRY-box transcription factor 17; *hnf4A*: Hepatocyte nuclear factor 4 alpha; *pdx1*: Pancreatic and duodenal homeobox 1; *sox9*: SRY-box transcription factor 9; *nkx6.1*: Nk6 homeobox 1 protein; *ngn3*: Neurogenin-3; *ins*: Insulin; *ptf1A*: Pancreas associated transcription factor 1A; *sst*: Somatostatin; *cgc*: Glucagon.

**Figure 4 ijms-25-09654-f004:**
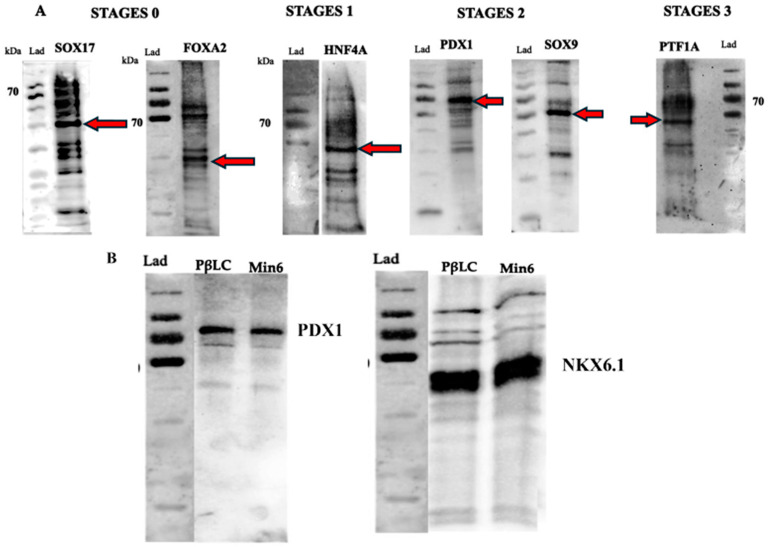
WB result of the differentiation process. The figure represents all the proteins detected during all the differentiation stages. (**A**) Proteins are detected individually at each differentiation stage. The approximate molecular weight of the protein (kDa) is shown. Red arrows showed the protein of interest. SOX17 44 kDa; FOXA2 49 kDa; HNF4A 52 kDa; PDX1 40 kDa; SOX9 63 kDa; PTF1A 42 kDa; NKX6.1 50 kDa. (**B**) Stage 4 of our differentiation process (PβLC). The results showed the expression of two characteristic proteins in a pancreatic-β cell (PDX1 and NKX6.1). The final results are compared to the insulin-secreting cells (Min6). Abbreviations: FOXA2: Forkhead box A2; SOX17: SRY-box transcription factor 17; HNF4A: Hepatocyte nuclear factor 4 alpha; PDX1: Pancreatic and duodenal homeobox 1; sox9: SRY-box transcription factor 9; nkx6.1: Nk6 homeobox 1 protein; ptf1A: Pancreas associated transcription factor 1A.

**Figure 5 ijms-25-09654-f005:**
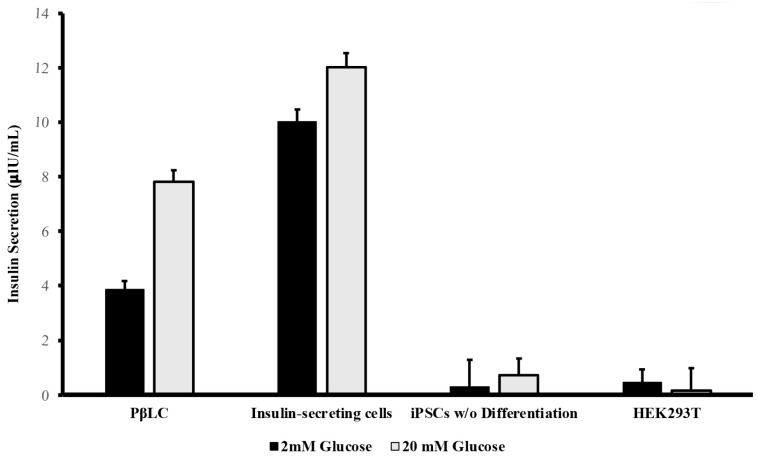
ELISA measurements of human insulin. Basal glucose was measured 1 h post-treatment with 2 mM (close Barr). Stimulated glucose was measured 1 h post-treatment with 20 mM glucose (open Barr). All measurements were performed in triplicate. PβLC: Pancreatic β-like cells; Insulin-secreting cells (Min6); iPSC w/o Differentiation: Induced pluripotent cells without differentiation. HEK293T: Human embryonic kidney cells.

**Figure 6 ijms-25-09654-f006:**
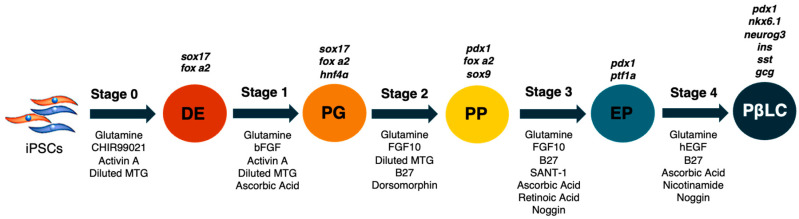
Representative scheme of the differentiation process. Abbreviations: DE: Definitive endoderm; PG: Pancreatic gut tube; PP: Pancreatic progenitor; EP: Endocrine progenitor; PβLC: Pancreatic beta-like cell. Gene expression is represented in black at the top of each differentiation stage. The arrows describe the cytokines and small molecules required to perform the protocol. Modified from Korytnikov and Nostro (2016) [21].

**Table 1 ijms-25-09654-t001:** Primer Utilized in qPCR.

Name	Forward	Reverse
*foxA2*	GGAACACCACTACGCCTTCAAC	AGTGCATCACCTGTTCGTAGGC
*sox17*	ACGCTTTCATGGTGTGGGCTAAG	GTCAGCGCCTTCCACGACTTG
*hnf4A*	GGTGTCCATACGCATCCTTGAC	AGCCGCTTGATCTTCCCTGGAT
*pdx1*	GAAGTCTACCAAAGCTCACGCG	GGAACTCCTTCTCCAGCTCTAG
*sox9*	AGGAAGCTCGCGGACCAGTAC	GGTGGTCCTTCTTGTGCTGCAC
*ptf1A*	GAAGGTCATCATCTGCCATCGG	CCTTGAGTTGTTTTTCATCAGTCC
*nkx6.1*	CCTATTCGTTGGGGATGACAGAG	TCTGTCTCCGAGTCCTGCTTCT
*ngn3*	CCTAAGAGCGAGTTGGCACTGA	AGTGCCGAGTTGAGGTTGTGCA
*ins*	ACGAGGCTTCTTCTACACACCC	TCCACAATGCCACGCTTCTGCA
*gcg*	CGTTCCCTTCAAGACACAGAGG	ACGCCTGGAGTCCAGATACTTG
*sst*	CCAGACTCCGTCAGTTTCTGCA	TTCCAGGGCATCATTCTCCGTC
*actB*	CACCATTGGCAATGAGCGGTTC	AGGTCTTTGCGGATGTCCACGT

## Data Availability

Data is contained within the article.

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
