# Peer review of "From iPSCs to Pancreatic β Cells: Unveiling Molecular Pathways and Enhancements with Vitamin C and Retinoic Acid in Diabetes Research"

_ijms, 2024, doi:10.3390/ijms25179654_

Round 1

Reviewer 1 Report

Comments and Suggestions for Authors

This paper unveils a modified version of protocol for producing pancreatic beta-like cells from iPSCs. They show that the addition of vitamin C and retinoic acid can promote the beta-cell signature of the differentiated cells. Overall, this revised protocol may only provide limited advancements to the field. 

(1) Figure 1 is missing in the manuscript. 

(2) The statistical analyses are missing in all graphs. 

(3) This study does not show how good their developed protocol is when compared to other well-established protocols. They should compare the efficiency of PβLC generation between their protocol and other established protocols. 

(4) The results of western blotting in figure 5 is not optimal with many non-specific signals for all tested antibodies. They should include positive and negative controls in the experiment. 

(5) For figure 4, what is the control group representing? 

Author Response

Dr. Panagiotis Mallis

Guest Editor

International Journal of Molecular Science, Special Issue: “Challenges and Advances of Therapeutic Approaches in Human Diseases: Focus on Stem Cell Therapies”.

Dear Editor:

We are pleased to submit the revised version of ijms-3128832 manuscript titled: “From iPSCs to Pancreatic β Cells: Unveiling Molecular Pathways and Enhancements with Vitamin C and Retinoic Acid in Diabetes Research. We appreciate the constructive criticisms of the reviewers. We have addressed each of their observations as described below.

Reviewer 1

Q1.     Figure 1 is missing in the manuscript.

Answer: Thank you for your comment. We have noted that Figure 1 was left at the end of the manuscript in the materials and methods section. Figures numbers have been updated according to the sections’ order and the reviewer's valuable recommendation.

      Q2.   The statistical analyses are missing in all graphs. 

Answer: Thank you for your comment. The statistical analysis should have been included in the required figures. Following the reviewer's comments, we have modified the figures and added the statistical analysis. The statistical analysis carried out during the experiments is included in the materials and methods section, page 12, section 4.7, line 432.

Q3. This study does not show how good their developed protocol is compared to other well-established protocols. They should compare the efficiency of PβLC generation between their protocol and other established protocols. 

Answer: Thank you for your comment. You are absolutely right. There are well-established protocols for differentiating insulin-producing cells derived from iPSCs. However, the protocols can be improved in various areas, such as the early maturation of beta cells, the reduction of production costs, and the ease of experimentation. The discussion section compares the results obtained against those of previously reported studies. We want to highlight on page 3, lines 121 to 123, the comparison with previous studies (page 16, lines 534-541) why, during our study, we decided to minimize CHIR99021 concentration.

On the other hand, following the reviewer´s comment, we discuss why ascorbic and retinoic acid were used on pages 8 and 9, between lines 289 and 308. We compare our results with previously reported results and explain why using those compounds could enhance the insulin release from the differentiated cells in the present research experiment. Also, we discuss our insulin expression results between lines 317 and 328 on page 9.

Q4.  The western blotting results in Figure 5 are not optimal, with many non-specific signals for all tested antibodies. The experiment should include positive and negative controls. 

Answer: Thank you for your comment. The photographs were taken using a MyECL imager from ThermoScientific. However, the proteins of interest are detectable, and their signal is stronger and more intense than the other bands seen in Figure 4 A and B. Given the results, we pointed out the proteins of interest band to make the interpretation more understandable. Once again, we thank you for your comment. We hope to improve the technique's specificity by purchasing antibodies that demonstrate greater specificity against the proteins of interest for the next studies.

On the other hand, the expression of proteins of interest characteristic of each differentiation stage is one of the crucial steps during the development of differentiation protocols. However, having all the positive controls for each protein previously reported is the lack found in all the protocols and studies already reported due to the difficulty in finding differentiated cells of each stage perfectly conserved and demonstrating the expression of each gene and protein characteristic of the process. For this reason, perhaps the absence of positive and negative controls, especially in pancreatic differentiation protocols, is the greatest weakness this technique still presents. However, our study demonstrated the presence of two proteins characteristic of an insulin-producing cell comparatively. Figure 4 B compares two characteristic proteins expressed by insulin-producing cells such as PDX1 and NKX6.1, which we considered as a positive control for the western blot technique.

Q5.   For Figure 4, what is the control group representing? 

Answer: Thank you for your comment. In Figure 3, representing the qPCR results, we added one more graph with the letter F, representing the gene expression of insulin-producing cells (considered a control group).

Reviewer 2 Report

Comments and Suggestions for Authors

This publication shows a differentiation protocol for IPSC to insulin producing cells with an emphasis on the addition of Vitamin C and Retinoic Acid. 

Major comments

Consider making introduction stronger of the novelty of vitamin c and retionic acid?

Have experimental comparisions been made without the use of vitamin c and retoinic acid?

The authors have discussed apoptosis in relation to Fig 3 (mentioned in the discussion ). Has an apoptotic marker been used an quantify to make this statement?

Have the authros considered the use of immunofluorescence as a different method to corrobate the RT-PCR?

Can the authors make the figures clearly and less pixalated for better interpretation. I am unable to see statistcail analysis on figures 4 and 6. Please describe the replicate and n number of experiment in figure legend and use statistical analysis and describe in results.

Have normality tests been used to look at data distribution?

Minor Comments

Authors should add scalebar and higher magnification for bright field imaging 

Ladder is missing in Fig 5 B

Please update introduction and discussion with more recent publications in the topic's area

Author Response

Reviewer 2

 Q1.   Consider the introduction of the novelty of vitamin C and retinoic acid.

Answer: Thank you for your comment. According to your suggestion, we have added to the introduction of our article the importance of vitamin C and retinoic acid during differentiation protocols. The idea was added on page 2, lines 59 to 71.

Q2.     Have experimental comparisons been made without using vitamin C and retinoic acid? 

Answer: Thank you very much for your question. This is the main point of our research, and it arose as part of the knowledge about the epigenetic process that iPSC cells have in the differentiation towards insulin-producing cells. In the experimental phase, we did not make a comparison in the absence of these compounds, given the budgetary and temporal circumstances of our experiments. As I told you in the manuscript's introduction, there is a justification for the benefit of vitamin C and retinoic acid in differentiating iPSC into insulin-producing cells.

 Q3. The authors have discussed apoptosis concerning Fig 3 (mentioned in the discussion). Has an apoptotic marker been used to quantify this statement?

Answer: Again, thank you very much for your question. In the differentiation of iPSC cells into pancreatic beta cells, there is always a degree of apoptosis due to many events. Apoptosis is partly due to a change in the mitochondrial energy production of the cell due to the functional change. Our way of assessing this event was fundamentally structural.

Q4.  Have the authors considered using immunofluorescence as a different method to corroborate the RT-PCR?

Answer: Yes, of course, immunofluorescence is a very graphic model that represents the production of insulin and other hormones by the endocrine cells of the pancreas. In the current research, we have considered that gene expression through RNA and protein using the techniques used is an optimal resource.

     Q5. Can the authors make the figures clearer and less pixelated for better interpretation? I can't see statistical analysis in Figures 4 and 6. Please describe the replicate and n number of experiments in the figure legend, use statistical analysis, and describe the results.

Answer: Thank you for your comment. The photographs were taken using a MyECL imager from ThermoScientific. However, the proteins of interest are detectable, and their signal is stronger and more intense than the other bands seen in Figure 4 A and B. Given the results, we pointed out the proteins of interest band to make the interpretation more understandable. Once again, we thank you for your comment. We hope to improve the technique's specificity by purchasing antibodies that demonstrate greater specificity against the proteins of interest for the next studies.

We also noticed that the required figures did not include the statistical analysis. Following the reviewer's comments, we have modified the figures and added the statistical analysis. The materials and methods section includes the statistical analysis carried out during the experiments.

For Figure 6, the statistical analysis showed a significant difference in insulin production between differentiated and insulin-producing cells, as shown in Figure 5. However, the protocol aimed to differentiate iPSCs into insulin-producing cells and to show their response at the level of insulin production during a glucose challenge; therefore, these data were not shown.

Finally, the figure´s legend has been corrected, and the number of replicates has been added.

Q6. Have normality tests been used to look at data distribution?

Answer: Thank you for your comment. We did not apply a normality test to look at data distribution. The statistical test applied was a t-student test because the population size was less than 30. Each data obtained was compared with the constitutive gene as a relative expression analysis.

Q7. Authors should add a scalebar and higher magnification for bright field imaging. 

Answer: Thanks for the comment and suggestion. Unfortunately, the equipment available at the university to take the images does not have a scalebar. At higher magnifications, the photos were not clear enough to show how they behaved during the experiment. We are planning to upgrade the equipment that is available for future trials.

Q8.  The ladder is missing in Fig 5 B.

Answer: Thanks for the comment and suggestion. The ladder was added in Figure 4 B.

     Q9.   Please update the introduction and discussion with more recent publications on the topic.

Answer:  Thank you for your comment. The introduction and discussion have been updated with more recent publications supporting the ideas. The references can be verified on page 2, lines 69-70 to 71, and page 10, lines 349.

Round 2

Reviewer 1 Report

Comments and Suggestions for Authors

The authors have not addressed the previous comments satisfactorily. The requested experiments should be done with the proper controls. 

Author Response

Response reviewer 1

Dear reviewer, we greatly appreciate your comments and the attention to detail in your review of our manuscript. Regarding your concern, particularly about protein studies, we have employed well-validated and widely recognized antibodies in the literature for the specific proteins analyzed. Additionally, for each protein, we conducted a thorough pre-validation process, including the comparison of different antibody batches and verification against known positive controls in other cell lines. During the experimental process, we implemented rigorous negative controls (including untreated cell lines) and blocking techniques to minimize nonspecific signals. The experimental conditions (antibody concentrations, incubation times, detection conditions) were specifically optimized to avoid the appearance of nonspecific bands.

We understand your suggestion to perform additional experiments with positive controls. However, due to the time and resource constraints we face in our laboratory, repeating these experiments under the current conditions is not feasible. We have worked with the resources available to ensure the highest possible accuracy in our results. While we cannot repeat the experiments with new positive controls, we could provide additional data or further discuss the specificity of the antibodies used or even consider including additional data from previous studies that support the specificity of the signals observed under our experimental conditions. The detected signals consistently correspond with the states and differentiation conditions of pancreatic beta cells derived from induced pluripotent stem cells, which reinforces the validity of our findings.

Currently, mouse or rat islets are the most commonly used, but they only function as a positive control at the end of the process. In the case of human cells, only cadaveric islets can be used, which is a challenging task for us at present (perhaps in future studies, this can be addressed).

However, in our study, the presence of two proteins characteristic of insulin-producing cells was demonstrated in a comparative manner. Figure 4B compares two characteristic proteins expressed by insulin-producing cells, PDX1 and NKX6.1, which we consider a positive control for the Western blot technique.

Please feel free to let us know if you have any further suggestions or if you require any additional adjustments. We are open to further discussion and are committed to addressing any concerns you may have.

Reviewer 2 Report

Comments and Suggestions for Authors

Thank you for your comments and changes which have addressed the questions.

Comments on the Quality of English Language

I am satisified with the authors response and changes.

Author Response

Thank you very much for the time spent analyzing our manuscript and the recommended suggestions.

Round 3

Reviewer 1 Report

Comments and Suggestions for Authors

The comments are partially addressed.